# Elucidating the Molecular Basis of Thermal Stress Response in Juvenile Turbot (*Scophthalmus maximus*) via an Integrative Transcriptome–Metabolome Approach

**DOI:** 10.3390/biology14101413

**Published:** 2025-10-14

**Authors:** Xiatian Chen, Tao Gao, Ziwen Wang, Shuaiyu Chen, Nan Zhang, Xiaoming Zhang, Yudong Jia

**Affiliations:** 1State Key Laboratory of Mariculture Biobreeding and Sustainable Goods, Yellow Sea Fisheries Research Institute, Chinese Academy of Fishery Sciences, Qingdao 266071, China; chenxiatian1216@163.com (X.C.); gaotao0098@hotmail.com (T.G.); 15038933884@163.com (Z.W.); shoushuaiyuchen@163.com (S.C.); 18526164783@163.com (N.Z.); 18766973648@163.com (X.Z.); 2Navigation and Ship Engineering College, Dalian Ocean University, Dalian 116023, China; 3National Engineering Research Center for Marine Aquaculture, Zhejiang Ocean University, Zhoushan 316022, China

**Keywords:** turbot (*Scophthalmus maximus*), thermal stress, gene expression, metabolite, correlation analysis

## Abstract

**Simple Summary:**

High temperatures can induce tissue damage and alter physiological processes in turbot (*Scophthalmus maximus*), yet the molecular mechanisms driving these changes remain largely unclear. In this study, we applied a combined transcriptomic and metabolomic approach to reveal the systemic responses of turbot under thermal stress. Our results revealed that heat stress leads to significant alterations in gene expression and metabolite profiles in the liver. Key genes were found to be closely linked to amino acid metabolism, particularly those involved in leucine, isoleucine, and valine pathways—as well as the metabolism of specific compounds such as galactonic acid. Pathway analysis further identified critical roles for the PI3K-Akt signaling pathway, protein transport, and protein processing in the thermal stress response. These pathways are essential for maintaining cellular homeostasis, regulating energy balance, and ensuring proper protein function under stress conditions. The findings deepen our understanding of the molecular basis of heat adaptation in turbot and may inform strategies to improve thermotolerance in farmed fish, with potential implications for sustainable aquaculture under rising climate pressures.

**Abstract:**

Temperature has always been an important environmental factor, and changes in water temperature are closely related to the entire life process of fish. Investigating the impact of thermal stress on fish physiology is critical for optimizing aquaculture. This study employed transcriptomic and metabolomic approaches to investigate temperature-induced variations in the gene expression and metabolic profiles of turbot. The results showed that thermal stress could induce abnormal genes transcription, and functional annotation demonstrated predominant associations of these genes with key pathways including PI3K-Akt signaling, PPAR regulation, steroid biosynthesis, fatty acid metabolism, and FoxO signaling cascade. Metabolomic analysis revealed that amino acid metabolism was dysregulated, such as valine, leucine, and isoleucine. Joint analysis revealed significant positive associations between *CDH1*, *Col9a1*, and *ECT2* genes and leucine/isoleucine metabolism. The expression levels of *Plch2* and *Col9a1* genes exhibited significant regulatory effects on valine metabolism. Moreover, the gene cluster comprising *DNAJB6*, *Gcnt1* and *trim71* was significantly involved in the metabolic regulation of galactonic acid. Collectively, these findings demonstrate that thermal stress induces significant alterations in gene expression, metabolic profiles, and signaling pathways in turbot, offering new perspectives for thermal stress mitigation strategies.

## 1. Introduction

The impact of increasing global seawater temperatures on marine life has become severe. Fish are ectotherms, meaning that their body temperature is primarily derived from the external environment. Consequently, the body temperature of most fish species conforms to that of the surrounding water. Their principal strategy for coping with temperature fluctuations is behavioral thermoregulation, such as moving between different water depths or habitats to find optimal temperatures. Many studies have reported that thermal stress can affect various aspects of a fish’s entire life process, including growth [1], oxidative response [2], immunity homeostasis [3] and epigenetic inheritance [4]. Under prolonged high-temperature conditions, Nile tilapia (*Oreochromis niloticus*) exhibited impaired growth performance and experienced stress-induced physiological damage [5]. Elevated temperatures affected both gill and skeletal muscle cellular morphology as well as immune regulation in black cuskeel (*Genypterus maculatus*), mediated through the upregulation of antibacterial peptides and pro-inflammatory cytokine expression [6]. Thermal stress can also alter the behavioral responses of fish, such as Atlantic salmon (*Salmo Salar*), including increasing inter-individual distances and modifying swimming depths [7]. Nevertheless, how elevated temperatures influence fish growth and development requires further in-depth research.

With the deepening understanding of nucleic acid molecules and the advancement of sequencing technologies, a pivotal methodology of RNA sequencing (RNA-seq) has been developed for transcriptome analysis and gene expression profiling [8]. Nowadays, RNA-seq methodologies enable the comprehensive investigation of diverse RNA biological features, including transcriptional dynamics, translational regulation, and structural characteristics [9,10,11]. Metabolomics is the analysis method of metabolites and is commonly used to discover the systemic effects of metabolites. In addition, owing to the high sensitivity of metabolomic approaches, even minor perturbations in biological pathways can be precisely identified, thereby providing mechanistic insights into diverse physiological states and pathological processes [12,13,14]. Integrated transcriptomic and metabolomic analyses enable the simultaneous investigation of biological systems from both gene regulation and metabolic response perspectives, which could facilitate the in-depth exploration of macroscopic developmental processes while elucidating the complexity and systemic integrity of biological phenomena [15,16,17]. Critically, this integrated approach is powerful for bridging the gap between genetic potential and phenotypic manifestation, allowing for the identification of key regulatory nodes that would remain hidden in single-omics studies.

The turbot (*Scophthalmus maximus*) is also called “Duobao fish” in China and is the major marine culture species in Northern China because of its high nutritional and economic value. At present, the breeding areas of turbot are mainly distributed in the northern coastal areas along the Bohai and Yellow Sea coasts. Our previous study has demonstrated that thermal stress could impair liver function in turbot by inducing apoptosis and injuring the hepatic capacity [18]. Yang et al. have found that high temperature stress could disturb the metabolic process in turbot, with a concomitant significant upregulation in creatinine and cortisol levels [19]. Zhao et al. reported significant alterations in hepatic lipid metabolism in turbot under thermal stress conditions [20]. However, the specific signaling pathways that are centrally involved, the coordinated interplay between gene expression and metabolic flux, and the key regulatory genes and metabolites that drive the thermal stress response in turbot remain largely unexplored and poorly integrated [21,22].

This study employed an integrated transcriptomic and metabolomic approach to elucidate the molecular mechanisms underlying the thermal stress response in turbot. By analyzing differentially expressed genes (DEGs) and metabolites, we aimed to precisely identify the core regulatory pathways and construct a gene–metabolite interaction network to uncover the systemic adaptive strategy of turbot under thermal stress. We expect that our findings will identify critical candidate genes and metabolic pathways that could serve as potential therapeutic targets for enhancing thermotolerance in turbot.

## 2. Materials and Methods

### 2.1. Study Design and Sample Collection

Healthy turbots were collected from Tianyuan Aquaculture Co., Ltd., Yantai, Shandong Province, China. In total, 120 fish (five months old) of similar weight (106.96 ± 15.71 g) and standard length (18.8 ± 0.9 cm) were randomly divided into four groups with three replicates per group and 10 fish per replicate (180 L/tank). The fish were acclimated in the tank for one week before the experiment and were fed a commercial pelleted diet twice a day until satiation at 9:00 and 15:00. Water quality parameters were measured and maintained at a salinity of 27‰–30‰, dissolved oxygen content was higher than 6.0 mg/L, pH was 7.5–8.0, and ammonia nitrogen content was less than 0.1 mg/L. Based on our previous results, we found that rapid warming can cause large-scale mortality of turbot [18]. The thermal acclimation protocol was implemented in a stepwise manner: first increasing from ambient 18 °C to 24 °C at 1 °C increments per 24 h period, then maintaining a slower escalation rate of 0.5 °C per 24 h period until the final test temperature of 27 °C was attained. The temperature was reduced from 27 °C to 18 °C at 1 °C per 1 h under flow-through conditions, with sampling conducted after 48 h of recovery at 18 °C. The controls were marked CTL, TA represented the group in 27 °C at 0 h, TB represented the group in 27 °C at 6 h, and TC represented the recovery group.

Following euthanasia with MS-222 (tricaine methane sulfonate, 100 mg/L; Sigma, St. Louis, MO, USA), hepatic tissues were aseptically excised using sterile surgical instruments and flash-frozen in liquid nitrogen [23].

### 2.2. RNA Isolation and Sequencing

Trizol reagent was used for RNA extraction from liver tissues. In brief, after tissue fragmentation with 1 mL of Trizol, 200 μL of chloroform was added, and centrifuged at 4 °C, 12,000 rpm, and 15 min. The supernatant was placed in a new tube with isopropanol added and then centrifuged at 4 °C, 12,000 rpm, and 10 min. RNA was dissolved in DEPC water. RNA quality was assessed using a NanoDrop (2000, Thermo Fisher Scientific, Waltham, MA, USA). mRNAs were enriched by magnetic beads and fragmented to 200–500 bp through the M220 Focused-ultrasonicator (Covaris M220, Covaris Inc., Waltham, MA, USA), using enzymes and random primers as templates to synthesize double-stranded cDNA, followed by repaired ends and adding the poly (A) and adapters. The sequencing was analyzed on the Illumina HiSeq platform of Gene Denovo Biotechnology (Guangzhou, China). The Trinity (https://github.com/trinityrnaseq (accessed on 10 May 2024)), BUSCO (http://busco.ezlab.org (accessed on 10 May 2024)), TransRate (http://hibberdlab.com/transrate/ (accessed on 10 May 2024)), and CD-HIT (http://weizhongli-lab.org/cd-hit/ (accessed on 10 May 2024)) platforms were used to reconstruct and evaluate the transcripts. Transcript abundance was quantified and normalized via the FPKM metric. The genes with fold change (FC) ≥ 2 and false discovery rate (FDR) < 0.05 were considered significant differences.

### 2.3. Gene Ontology and KEGG Analysis

The Gene Ontology (GO) database was used to classify and annotate DEGs according to their participation in molecular function, biological process and cellular component. KEGG function analysis was performed by KOBAS (http://bioinfo.org/kobas/ (accessed on 10 May 2024)).

### 2.4. Metabolites Extraction and Analysis

The liver samples used for this metabolomic analysis were collected from the same individuals as those used for transcriptomic profiling. Liver tissue samples of approximately 50 mg were homogenized in 1.5 mL centrifuge tubes with a precise volume of L-2-chlorophenylalanine (as an internal standard) and methanol. The mixture was incubated at −20 °C for 10 min and mechanically homogenized. Subsequently, it was subjected to ultra-sonication in an ice-water bath for 10 min, followed by incubation at −20 °C for 30 min. The samples were then centrifuged at 13,000× *g* for 10 min at 4 °C. The resulting supernatant was transferred to a new vial and lyophilized in a vacuum concentrator at ambient temperature. The dried extract was reconstituted in an appropriate volume of methanol and centrifuged again under the same conditions (13,000× *g*, 10 min, 4 °C), and the final supernatant was injected into a gas chromatography-mass spectrometry system (GeneDenovo Biotechnology, Guangzhou, China). Data acquisition was performed using Chroma TOF 4.3X software (LECO), and metabolites were filtered with a quality control criterion of relative standard deviation < 30% in the quality control samples.

### 2.5. Pathway Analysis

The differentially expressed metabolites were annotated on the established biochemical databases (KEGG and PubChem). Subsequently, the metabolic pathways were mapped in the database of corresponding species (*Cynoglossus semilaevis*) on the MetaboAnalyst website.

### 2.6. Association Analysis of Metabolome and Transcriptome

Two-way orthogonal partial least squares (O2PLS) is a generalized OPLS that can be used for bidirectional modeling and prediction in two data matrices. With this analysis, the internal connections between the two omics can be explored, the degree of association between the two omics data can be determined, and the main genes, metabolites, or proteins that cause this association can be identified. The DEGs-metabolites network was performed on the OmicShare platform (https://www.omicshare.com/tools/ (accessed on 17 May 2024), Gene Denovo) [24].

### 2.7. Real-Time Quantitative PCR (qPCR)

One μg RNA was reversed into cDNA using the cDNA synthetis kit (Cat#R323, Vazyme, Nanjing, China). The expression of genes was detected by the Real-time Quantitative Thermal Cycler (MA-6000, Molarray, Suzhou Molarray Biological Technology Co., Ltd., Suzhou, China), and SYBR Green reagent (Cat#R711, Vazyme, Nanjing, China) was used for progress tracking. The cycler reaction was composed of 1 × SYBR Green mix, 200 nM primer (forward/reverse), and 10 ng cDNA template, with nuclease-free water added to 20 μL volume. The thermal protocol was as follows: 95 °C for 30 s, followed by 40 cycles (including 95 °C for 10 s and 60 °C for 30 s) and then 95 °C for 15 s, 60 °C for 1 min, and 95 °C for 15 s. The data were analyzed using the 2^−ΔΔCt^ method [18,25]. *GAPDH* was set as an internal reference gene. All primer pairs were designed with Primer 3 software (the sequences are provided in Table 1).

### 2.8. Statistical Analysis

All data were shown as mean ± standard deviation. After verifying assumptions of normality and the homogeneity of variances, inter-group differences were assessed by one-way ANOVA. Post hoc comparisons were made using the least significant difference test, with findings further validated by Tukey’s test. Statistical analyses were performed with IBM SPSS Statistics (Version 27.0.1.0). *p* < 0.05 was considered a significant difference between the groups.

## 3. Results

### 3.1. Transcriptome Sequencing and Data Analysis

After comparing and analyzing the sequencing results, a total of approximately 85.83 Gb bases and 581.23 Mb reads were obtained from 12 libraries. The average data of sequencing quality to Q20 and Q30 were 98.37% and 94.92%, respectively. The average of GC content was 54.52% (Table 2). The distribution of gene expression abundance showed a normal distribution of gene expression in each group of samples (Figure 1A). Correlation analysis revealed significant divergence between control and experimental groups, with the TA group exhibiting particularly distinct clustering patterns (Figure 1B). These results indicate that the quality of sequencing was verified, and the data were reliable.

### 3.2. Differentially Expressed Genes Under High Temperatures in Turbot

Overall, 5702 genes were differently expressed in the three groups. Among them, there were 3070, 193, and 171 upregulated mRNA and 1745, 405, and 118 downregulated mRNA in the CTL vs. TA, CTL vs. TB, and CTL vs. TC groups, respectively (Figure 2A–D, Appendix A). The top 10 DEGs of the three groups are shown in Table 3, Table 4 and Table 5. Venn diagram analysis showed that there are 395 common genes between the CTL vs. TA group and the CTL vs. TB group, 170 common genes between the CTL vs. TB group and the CTL vs. TC group, 162 common genes between the CTL vs. TC group and the CTL vs. TA group, and 126 common genes among the three groups (Figure 2E).

### 3.3. Enrichment Analysis of DEGs

In the CTL vs. TA group, the DEGs were mainly enriched in the metabolic process, localization, immune system process, biological adhesion, single-organism process, cellular component organization or biogenesis, and developmental process in the BP category. In the CC category, the DEGs were associated with the membrane part, organelle part, cell part, and macromolecular complex. In the MF category, the DEGs were involved in molecular signal transduction, catalytic activity, and binding transcription factor activity (Figure 3A, Appendix A).

The DEGs in the CTL vs. TB group were mainly involved in the cellular process, biological regulation, metabolic process, and response to stimulus in the BP category. Moreover, those genes were involved in the cell, cell part, organelle, and membrane processes in the CC part. In the MF category, the DEGs were significantly involved in catalytic activity and complex binding (Figure 3B, Appendix A). In the CTL vs. TC group, the process of those dysregulated genes was similar to the above group, such as molecular signal transduction, catalytic activity, stimulus response and binding (Figure 3C, Appendix A).

KEGG analysis results revealed that the DEGs in the CTL vs. TA group were mainly correlated with protein processing, arginine and proline metabolism, the Toll signaling pathway, the TCA cycle, the PPAR signaling pathway, metabolism processes and apoptosis (Figure 3D). In the CTL vs. TB group, the dysregulated genes were associated with cell cycle, DNA replication, steroid biosynthesis, chemokine signaling pathway, amino acids biosynthesis, the PPAR signaling pathway, ECM-receptor interaction, and fatty acid metabolism (Figure 3E). In the CTL vs. TC. group, the dysregulated genes were associated with steroid biosynthesis, autophagy regulation, the PPAR signaling pathway, adipocytokine signaling, fatty acid metabolism and FoxO signaling pathway (Figure 3F).

### 3.4. Differential Metabolites Under High Temperatures in Turbot

The PCA analysis exhibited that there was significant distribution among the samples (Figure 4A). The volcano plot showed that 64 metabolites were detected in the CTL vs. TA group, of which 27 were upregulated, and 37 were downregulated. There were 56 differential metabolites in the CTL vs. TB group, of which 37 were upregulated and 19 were downregulated. Moreover, there were 69 differential metabolites in the CTL vs. TC group, of which 30 were upregulated and 39 were downregulated (Figure 4B–D). The metabolome results showed that the most dysregulated metabolites associated with high temperature stress included 4-hydroxy-3-methoxybenzoic acid, 2-amino-3-methoxybenzoic acid, 1, 2, 3-dimethylsuccinic acid, lipoic acid, and L-glutamic acid (Appendix A). However, the most dysregulated metabolites were trans-4-hydroxy-L-proline 1, 3-hydroxy-3-methylglutaric acid, lactobionic acid 1, α-santonin 1, and methyl palmitoleate in the CTL vs. TC group (Appendix A). Of particular interest, we detected a signal in the liver metabolome of turbot that corresponds to α-santonin 1. Since this compound is unlikely to be of endogenous origin, the observed signal may represent an as-yet-unidentified endogenous metabolite sharing structural or chemical similarities with α-santonin. Alternatively, it could have been derived from plant-based components in the formulated diet and subsequently accumulated in the liver through dietary intake.

### 3.5. Pathway Analysis of Metabolites

The results of pathway analysis showed that although there were different metabolites between the high temperature and recovery groups, these metabolites were involved in the same pathways, such as amino acid biosynthesis and metabolism (e.g., valine, leucine, alanine, aspartate), glyoxylate and dicarboxylate metabolism, glycerolipid metabolism, and the TCA cycle (Figure 5, Appendix A).

### 3.6. Integrated Transcriptomic and Metabolomic Profiling

Thirty dysregulated genes and 27 metabolites were loaded in the analysis, including *DNAJB6*, *CDH1*, *TDH*, *slc25a25a*, *CDHR2*, *PCK1*, leucine, valine, galactonic acid and isoleucine (Figure 6A,B). Integrated analysis identified significant correlations between DEGs and metabolites. As shown in Figure 6C, valine, isoleucine, leucine, galactonic acid, guanine 2, and methionine 1 showed higher correlation with DEGs.

Moreover, we analyzed the correlations between these differential metabolites and differential genes and the correlation coefficient absolute values of >0.5 were plotted on the heat map (Figure 6D, Appendix A). The four downregulated genes (*Plch2*, *CISH*, *Slco2a1*, *ECT2*) were negatively correlated with the 13 upregulated metabolites, including 4-aminolhenol 1, valine, isoleucine, leucine, L-cysteine, 2,3-dimethylsuccinic acid, and methionine 1. Correlation analysis revealed significant positive associations between *CDH1*, *Col9a1*, and *ECT2* genes and leucine/isoleucine metabolism. The expression levels of *Plch2* and *Col9a1* genes exhibited significant regulatory effects on valine metabolic pathways. Moreover, *CISH* and *Plch2* genes showed strong correlations (*r* > 0.8) with leucine and isoleucine metabolism, while the gene cluster comprising *DNAJB6*, *Gcnt1*, and *trim71* was significantly involved in the metabolic regulation of galactonic acid.

### 3.7. Identification of the Expression of DEGs by qPCR

The expression of 10 dysregulated key genes correlated with the metabolism was validated in the liver of turbot (Figure 7). The expression of *TDH*, *Hsp90a.1*, *slc25a25a*, *DNAJB6*, and *Hsp70* in the liver was higher than in the control under the heat stress, and *CYP7A1*, *GNRHR2*, *NPR2*, *CASP6* and *GCK* were downregulated. The results were consistent with the transcriptomic sequencing data. However, in the recovery group, the expression of most genes, except for *slc25a25a*, *DNAJB6*, and *CYP7A1*, still showed significant differences compared to the control group.

## 4. Discussion

As a predominant abiotic stressor, thermal elevation beyond optimal ranges disrupts homeostasis in cold-water fish, triggering cascading effects from molecular to organismal levels. Elucidating the molecular mechanisms underlying thermal stress responses is fundamental for developing conservation strategies to mitigate climate change impacts on fisheries. In this study, we identified many DEGs by using transcriptome sequencing. A total of 5413 genes were dysregulated in fish under heat stress, including 126 common genes, which can provide targets for subsequent research on resistance to high temperature stress. We found that fibulin-7 (*FBLN7*) was downregulated in thermal stress and exhibited the highest differential expression at 27 °C for 0 h. *FBLN7* is an extracellular matrix (ECM) adhesion protein that mediates interactions with various ECM components, including structural proteins, cell surface receptors, and growth factors [26]. Sarangi et al. found that it can regulate the migration and infiltration of monocytes and macrophages and reduce the expression of inflammatory factors [27]. We considered that *FBLN7* may participate in regulating the occurrence of inflammation in fish under high temperature stress. During the stress process, animals break down their own tissues to generate energy, which is directed toward specific tissues while also reducing the energy supplied to other tissues to resist damage caused by stress stimulation [28,29,30].

*GCK* plays a vital role in intracellular glucose uptake and utilization, not only initiating all major pathways of glucose utilization but also maintaining the gradient concentration required to promote glucose entry into cells [31,32]. The expression of *GCK* was downregulated in the liver of turbot under heat stress, according to the results from qPCR and transcriptome analysis. *SET-1* is a class of epigenetic modification enzymes containing SETD, mainly affecting gene expression by modifying the methylation of histones *H3K4*, *H3K9*, *H3K36*, and *H4K20* [33]. SETD can also catalyze the methylation of non-histones, thereby affecting signal transduction and participating in DNA repair and other processes [34,35,36]. In the present study, *SET-1* expression was among the top 10 upregulated genes in turbot under thermal stress, suggesting a potentially high level of methylation activity. Epigenetic regulation—including histone modification and DNA methylation—has been increasingly recognized as a key mechanism in fish stress responses, with heat-induced changes reported in species such as zebrafish and salmon [37,38]. In fish, temperature stress can influence membrane fluidity, gene expression, and metabolic processes, and epigenetic mechanisms like those mediated by *SET-1* are thought to help orchestrate these adaptations. Potential approaches, such as ChIP-seq to assess genome-wide histone methylation patterns or DNA methylation assay, are needed to clarify whether *SET-1*-mediated methylation directly influences stress-responsive gene expression in turbot. Moreover, research in other models indicates that SET1 protein levels can be regulated by factors such as the APC/C^Cdh1^ complex and Cla4 kinase [39], suggesting that the post-translational regulation of *SET-1* may also be important in fish’s thermal response.

Amino acids are the basic building blocks of biological functional macromolecular proteins [40,41]. They are raw materials used in the body to manufacture antibody proteins, hemoglobin, enzyme proteins, hormone proteins, and neurotransmitter substances and can even be used to provide energy sources for living organisms. In this study, we found that the metabolic processes were accelerated under high temperatures, including the biosynthesis of amino acids, the TCA cycle process, CoA synthesis, and lipid metabolism. Steroids are considered to be the analog of glucocorticoids that can effectively activate the body’s immune defense system [42,43]. Numerous studies have shown that steroids can inhibit intracellular mediated immune responses, and the signaling of steroids is activated to maintain the body’s homeostasis [44,45,46]. The pathway of steroid biosynthesis was significantly enriched in turbot under heat stress, which was similar to the previous studies. Cheng et al. reported that supplementing steroids with largemouth bass can effectively alleviate heat stress-induced immune damage and metabolic disorders [47]. Moreover, Zhang et al. found that the synthesis of steroids was impaired under salinity stress, accompanied by a decrease in reproductive and immune function and an increased risk of infection [48].

The body will increase its metabolic response to resist the influence caused by external pressure stimulation, and then, the respiratory metabolism of mitochondria will be accelerated, resulting in significant ROS generation and insufficient clearance, resulting in the generation of oxidative stress [49,50,51]. Moreover, oxidative stress was associated with the apoptosis process. In the previous study, we found that thermal stress could induce the apoptosis of hepatic cells in turbot [18]. Liu et al. reported that thermal stress significantly alters the activity profile of hepatic antioxidant enzymes (e.g., SOD, CAT) while initiating programmed cell death pathways in Clarias fuscus liver tissue [52]. In this study, we found that the dysregulated genes and metabolites were related to the process of oxidative stress, autophagy, and apoptosis, including the PPAR signaling pathway, amino metabolic, biosynthesis, and the FoxO signaling pathway. PPAR is the nuclear receptor involved in the regulation of energy metabolism, cell development and differentiation [53,54,55]. In addition, Li et al. revealed that PPAR participated in lipid metabolism and the response to stress stimulation during the embryonic development of fish [56]. Specifically, the PPAR pathway could help maintain energy homeostasis by regulating fatty acid oxidation and lipid metabolism, a conserved mechanism observed in species like large yellow croaker (*Larimichthys crocea*) and Nile tilapia (*Oreochromis niloticus*) under temperature fluctuations [57,58]. Forkhead box proteins represent a conserved family of transcription factors that regulate apoptotic pathways across species by upregulating key pro-apoptotic factors, including Dally-like protein, Bcl-2 antagonist/killer, and tumor necrosis factor-related apoptosis-inducing ligand [59]. The FoxO signaling pathway detected here is a central regulator of cellular stress response, promoting the expression of antioxidant enzymes (e.g., SOD, CAT) to mitigate oxidative damage and inducing autophagy to maintain cellular homeostasis, as similar studies in zebrafish (*Danio rerio*) and common carp (*Cyprinus carpio*) reveal [60,61,62]. These results showed that when fish underwent the thermal stress, apoptosis and inflammatory pathways were mainly activated and enriched.

Through associated analysis of transcriptome and metabolome, it was found that some genes and metabolites exhibit correlations under high temperature stress, such as *TDH*, *slc25a25a*, *Plch1*, *CISH*, and *PCK1*. Moreover, we selected and validated the expression of those key genes in turbot liver, and the results were consistent with the sequencing. Correlation analysis revealed that genes (e.g., *COL9A1*, *PCK1*, *SGK1*, and *DNAJB6*) were significantly associated with the metabolism of leucine, galactonic acid, isoleucine, and valine, suggesting their potential regulatory roles in amino acid metabolism under high temperature stress. Furthermore, these DEGs (e.g., *COL9A1*, *PCK1*, *SGK1*, *DNAJB6*) were primarily enriched in intracellular protein folding, nuclear transport, and the PI3K-Akt signaling pathway. The PI3K-Akt pathway serves as a critical hub coordinating cell survival, growth, and metabolism. Its activation under heat stress likely represents a protective mechanism to inhibit apoptosis and support energy-demanding repair processes. These findings imply that the above signaling pathway may mediate the physiological response of turbot to thermal stress. Collectively, the enrichment of the PPAR, FoxO, and PI3K-Akt pathways highlights a coordinated transcriptional and metabolic reprogramming in turbot liver aimed at balancing energy allocation, mitigating oxidative injury, and determining cell fate under thermal stress. While these core pathways are evolutionarily conserved, the specific gene–metabolite interactions identified here, such as the strong association between *COL9A1* and branched-chain amino acid metabolism, may represent species-specific regulatory features in turbot’s adaptation to high temperature.

## 5. Conclusions

In summary, the synergistic application of multi-omics approaches has elucidated the systemic responses of turbot to thermal stress, identifying coordinated dysregulation of transcriptional networks and metabolic pathways critical for thermal tolerance. Moreover, joint analysis revealed significant positive associations between *CDH1*, *Col9a1*, *and ECT2* genes and leucine/isoleucine metabolism. The expression levels of *Plch2* and *Col9a1* genes exhibited significant regulatory effects on valine metabolism. *CISH* and *Plch2* genes showed strong correlations with leucine and isoleucine, while the gene cluster comprising *DNAJB6*, *Gcnt1*, and *trim71* was significantly involved in the metabolic regulation of galactonic acid. These dysregulated genes and metabolites were mainly involved in ABC transporters, the PI3K-Akt signaling pathway, and protein transport and processing. The identified biomarkers could guide the selective breeding of thermotolerant fish, while the metabolic insights inform the development of targeted management practices to mitigate heat stress, thereby enhancing the industry’s resilience to rising temperatures. These findings offer valuable insights for addressing environmental stressors in turbot aquaculture. However, it should be noted that as the fish in this study were sourced from a commercial processing company, the potential genetic diversity within the population may introduce variability in thermal response, which represents a limitation of the present study. Thus, further validation and multi-generational studies are required to validate the findings of this study.

## Figures and Tables

**Figure 1 biology-14-01413-f001:**
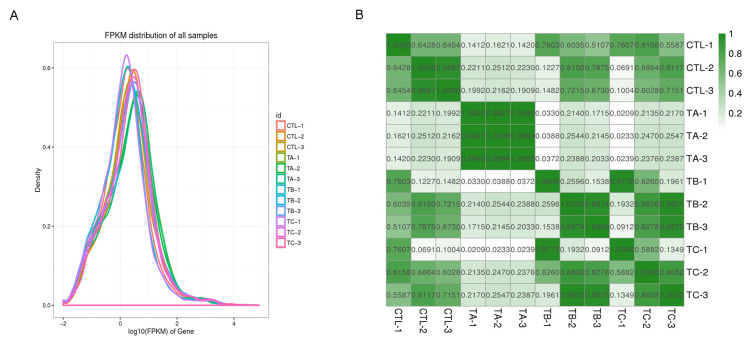
The distribution and correction analysis of each sample. (**A**) The FPKM distribution of samples. The X axis is the log_10_ FPKM of the gene. The Y axis is the density. (**B**) Each color block represents the correlation between X and Y samples. The depth of the color represents the high correlation, and the number in the color represents the correlation value.

**Figure 2 biology-14-01413-f002:**
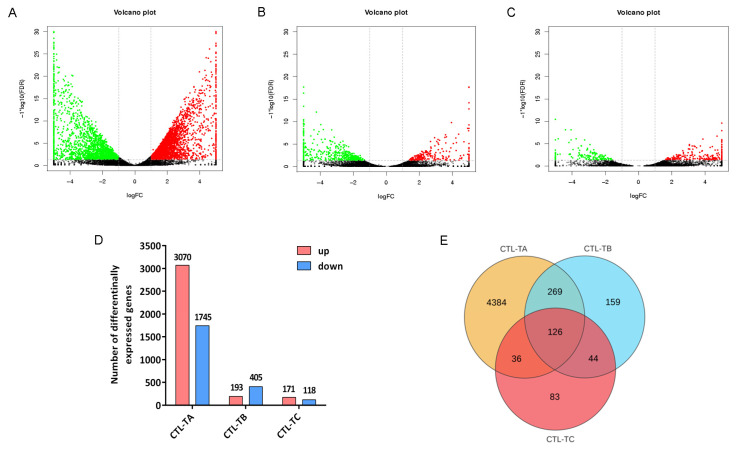
Sequencing profiles of DEGs in each group. Volcano plots show DEGs in (**A**) CTL vs. TA group, (**B**) CTL vs. TB group, and (**C**) CTL vs. TC group. The X axis represents the log_2_FC, and the Y axis represents the −log_10_ value of the FDR. The red, green and black dots represent upregulated genes, downregulated genes, and no significant genes, respectively. (**D**) The number of dysregulated genes in the three groups. The numbers on the bars represent the number of genes. (**E**) Venn diagram of genes among three comparisons.

**Figure 3 biology-14-01413-f003:**
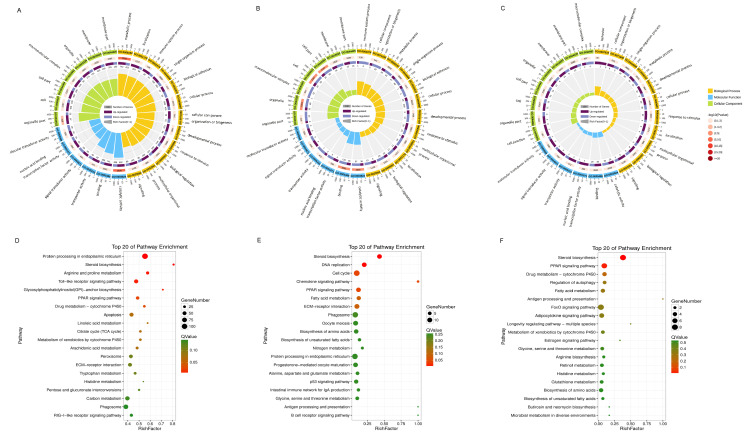
The enrichment analysis of dysregulated genes. GO analysis of dysregulated genes in (**A**) CTL vs. TA group, (**B**) CTL vs. TB group, (**C**) CTL vs. TC group. The outside circle represents the class, and each color represents a class; the ruler represents the number of genes. The second circle represents the number of genes in this pathway and the *p* value. Longer lines mean more genes, and darker colors mean a smaller *p* value. The third circle shows the dysregulated genes in this group. The fourth circle shows rich factor values, and the larger form indicates the higher the degree of enrichment. KEGG analysis of DEGs in (**D**) CTL vs. TA group, (**E**) CTL vs. TB group, and (**F**) CTL vs. TC group.

**Figure 4 biology-14-01413-f004:**
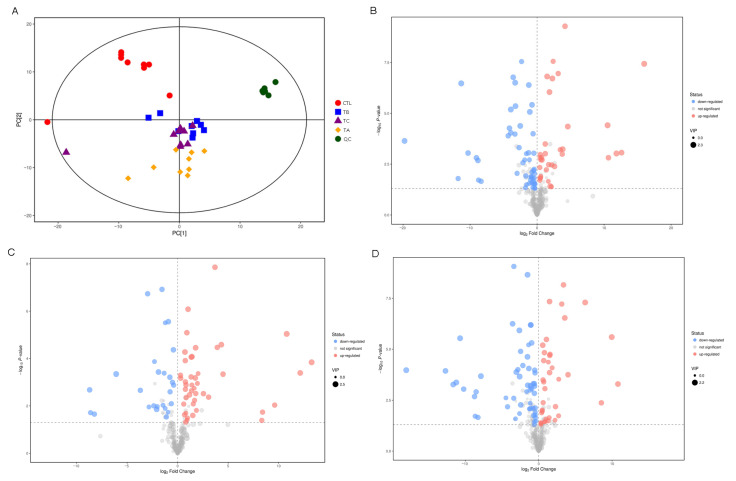
Differential metabolites in the samples. (**A**) PCA analysis of the sample distribution. Each dot represents one sample. The volcano plot shows the metabolites in (**B**) CTL vs. TA group, (**C**) CTL vs. TB group, and (**D**) CTL vs. TC group. The X axis represents the log_2_ FC, and the Y axis represents the log_10_
*p*-value. Red dots indicate upregulated metabolites and blue dots indicate downregulated metabolites.

**Figure 5 biology-14-01413-f005:**
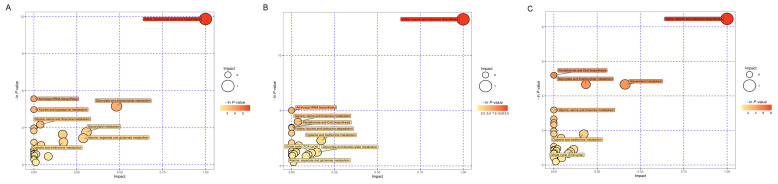
Pathway analysis of metabolites. The pathway of metabolites in (**A**) CTL vs. TA group, (**B**) CTL vs. TB group, and (**C**) CTL vs. TC group. Each bubble indicates a pathway. The X axis indicates the impact, and the Y axis indicates the log_10_
*p*-value.

**Figure 6 biology-14-01413-f006:**
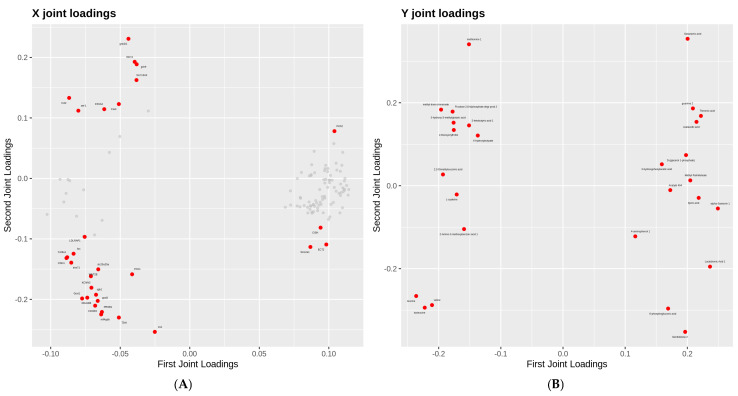
Transcriptome and metabolome associated analysis. (**A**,**B**) O2PLS load diagram. X joint loadings represent the transcriptome data and Y joint loadings represent the metabolome data. The dots indicate genes or metabolites, and the absolute value indicates the degree of association between the two elements. (**C**) Bipartite load diagram. The red dot indicates the gene and the triangle indicates the metabolite. (**D**) Heat map of correlation between key genes and metabolites. The X axis indicates the metabolites, and the Y axis indicates the gene. Red indicates a positive correlation, while blue indicates a negative correlation. The asterisks indicate the degree of correlation. The more the asterisks, the higher the correlation.

**Figure 7 biology-14-01413-f007:**
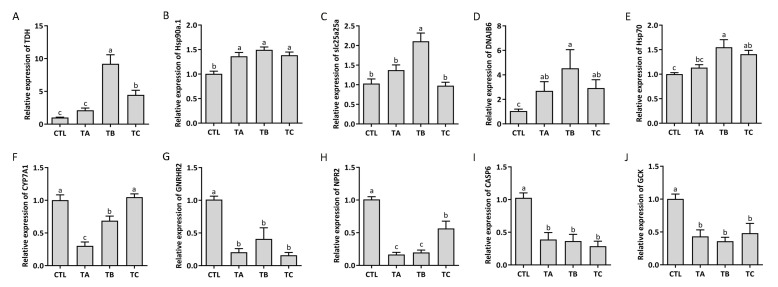
The expression levels of key genes under thermal stress in turbot. (**A**) *TDH* expression. (**B**) *Hsp90a.1* expression. (**C**) *slc25a25a* expression. (**D**) *DNAJB6* expression. (**E**) *Hsp70* expression. (**F**) *CYP7A1* expression. (**G**) *GNRHR2* expression. (**H**) *NPR2* expression. (**I**) *CASP6* expression. (**J**) *GCK* expression. The letter above indicates the difference. Different letters represent the differences between the groups.

**Table 1 biology-14-01413-t001:** Characteristics of primers in this study.

Gene Name	Primer	Sequences	Tm (°C)
*TDH*	Forward	CCATCCCAAGGTGCTCATCA	61.2
	Reverse	CCGCAGGTTCTTGTAGTCCA	58.5
*hsp90a.1*	Forward	GGAGTTCGACGGTAAGAGCC	58.8
	Reverse	AGTCTGTTGGACACCGTCAC	54.8
*hsp70*	Forward	CGCCCTGGAGTCGTATGCTTTC	65.8
	Reverse	GCCAGCCGATGACCTCGTTGC	69.4
*slc25a25a*	Forward	TGCTGAAAACTCGTCTCGCT	58.5
	Reverse	CATGTTGGGGACGTAGCCTT	59.8
*DNAJB6*	Forward	TTACGGGGGCTTTGTTGGTT	61.9
	Reverse	GGAGAAAGAGGTGAAGCCCC	60.1
*CYP7A1*	Forward	GGGCAGTAGTGGTGGGATTC	59.0
	Reverse	TGCCCATACTTCTTCTGCCG	61.1
*GNRHR2*	Forward	GCTGGGGCTGCTTCTATGTG	60.6
	Reverse	CTGAAAGAGTCCAGCTCCCTC	58.1
*NPR2*	Forward	GTGTGGTGGACAGTCGATTTG	58.3
	Reverse	TGGAGAACTTGCATAGAGTGCG	60.6
*CASP6*	Forward	ATCGGGGTTGTCTTGTCGAA	60.1
	Reverse	GCTGCATTTCCCAGCACTTC	60.6
*GCK*	Forward	TTTTGGTCGCGTTGAGTTCTG	61.3
	Reverse	CGGCGCAGTTAGACGAGAAA	61.5
*GAPDH*	Forward	AGTCCGTCTGGAGAAACCC	57.0
	Reverse	CAAAGATGGAGGAGTGAGTGT	54.1

**Table 2 biology-14-01413-t002:** Transcriptome sequencing data of each group.

Group	Sample	Clean Data (bp)	Clean Reads (Mb)	Q20 (%)	Q30 (%)	GC Content (%)
CTL	CTL-1	6,475,702,062	43.853940	98.36%	94.89%	55.15%
	CTL-2	7,099,878,863	48.167156	98.37%	94.98%	54.07%
	CTL-3	7,358,091,696	49.814014	98.44%	95.12%	53.93%
TA	TA-1	6,361,078,072	43.014726	98.41%	95.03%	53.96%
	TA-2	7,105,868,744	48.142580	98.38%	95.00%	53.68%
	TA-3	7,545,678,321	51.070376	98.45%	95.16%	53.28%
TB	TB-1	7,910,701,740	53.563198	98.29%	94.68%	56.63%
	TB-2	7,118,001,549	48.180086	98.36%	94.91%	54.20%
	TB-3	7,705,624,944	52.164258	98.37%	94.93%	54.32%
TC	TC-1	6,901,596,551	46.778486	98.31%	94.75%	56.07%
	TC-2	6,732,537,497	45.604702	98.30%	94.76%	54.71%
	TC-3	7,515,726,757	50.878070	98.35%	94.88%	54.19%

**Table 3 biology-14-01413-t003:** The top 10 DEGs in the CTL vs. TA group.

ID	Symbol	Log_2_ (FC)	*p* Value	Expression
SMAX5B002285	*RBP7*	13.39026	8.46 × 10^−9^	up
SMAX5B008965	*HHIPL2*	12.17374	4.64 × 10^−9^	up
SMAX5B006145	*MYOC*	12.09452	3.35 × 10^−15^	up
SMAX5B021851	*CRYGNB*	11.8778	4.13 × 10^−11^	up
SMAX5B018082	*NEFM*	11.5134	2.53 × 10^−7^	up
SMAX5B007292	*EPD*	−16.70846	1.83 × 10^−9^	down
SMAX5B012667	*FBLN7*	−15.29945	6.85 × 10^−30^	down
SMAX5B018368	*RDH7*	−15.01527	7.17 × 10^−51^	down
SMAX5B015453	*CYB5R2*	−14.81061	5.02 × 10^−34^	down
SMAX5B008939	*FAM213B*	−14.77297	1.58 × 10^−50^	down

**Table 4 biology-14-01413-t004:** The top 10 DEGs in the CTL vs. TB group.

ID	Symbol	Log_2_ (FC)	*p* Value	Expression
SMAX5B018187	*gcg2*	14.85200924	2.62 × 10^−5^	up
SMAX5B012775	*hsp30*	11.63026713	3.80 × 10^−6^	up
SMAX5B022597	*dio1*	10.86108691	9.79 × 10^−5^	up
SMAX5B010135	*set-1*	10.43045255	3.76 × 10^−7^	up
SMAX5B011509	*LRAT*	10.38801729	4.48 × 10^−5^	up
SMAX5B002496	*ENDOD1*	−10.81645038	7.52 × 10^−8^	down
SMAX5B020931	*Tmem235*	−10.79495732	3.01 × 10^−5^	down
SMAX5B012680	*Cenpq*	−10.6911619	7.00 × 10^−6^	down
XLOC_024262	*Vcam1*	−10.55074679	0.000306632	down
SMAX5B022022	*Irs2*	−10.51175265	0.000106176	down

**Table 5 biology-14-01413-t005:** The top 10 DEGs in the CTL vs. TC group.

ID	Symbol	Log_2_ (FC)	*p* Value	Expression
SMAX5B018187	*gcg2*	18.60534	2.47 × 10^−9^	up
SMAX5B017929	*Pcsk2*	11.18818	1.57 × 10^−6^	up
SMAX5B000021	*SCG5*	11.14636	3.14 × 10^−6^	up
SMAX5B011417	*baiap2l2*	10.99906	4.59 × 10^−9^	up
SMAX5B010135	*set-1*	10.5216	1.60 × 10^−6^	up
SMAX5B020931	*Tmem235*	−10.79495732	5.25 × 10^−5^	down
SMAX5B022022	*Irs2*	−10.51175265	0.000232012	down
SMAX5B009464	*tuba*	−10.22480563	0.000216174	down
XLOC_008456	*Myl7*	−9.936637939	0.000533848	down
SMAX5B019138	*Msn*	−9.631783357	0.000131621	down

## Data Availability

The original contributions presented in the study are included in the article, further inquiries can be directed to the corresponding author. The data used to support the findings of this study have been deposited in the NCBI’s Sequence Read Archive (SRA), reference number (BioProject ID: PRJNA1344168). qPCR data have been made publicly available via the Zenodo repository (https://doi.org/10.5281/zenodo.17299068).

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
