# Peer review of "Elucidating the Molecular Basis of Thermal Stress Response in Juvenile Turbot (Scophthalmus maximus) via an Integrative Transcriptome–Metabolome Approach"

_biology, 2025, doi:10.3390/biology14101413_

Round 1

Reviewer 1 Report

Comments and Suggestions for Authors

Dear Authors and Editors,  

The manuscript presents new data obtained by ntegrative transcriptome-metabolome approach and is important for studying the molecular basis of thermal stress response in fish in general and turbot in particular.

The cutting-edge integration of metabolomics (GC-MS) and transcriptomics (RNA-seq) data allows us not only to document changes but also to reconstruct a holistic picture of the biochemical response to stress, establishing causal relationships between gene expression and metabolic changes. Their connection consists of complementary levels of information.

The authors logically chose the tissue for analysis. Complex metabolic changes occur in the liver during heat stress. GC-MS analysis allows for a detailed characterization of these changes, providing valuable information about the physiological state of fish and their adaptation mechanisms to stress.

The manuscript is worth publishing, but after some revision.

There are several comments, which I think should be considered before publication in the journal:

Materials and Methods:

Experiment Design and Sample Collection are not clearly described. You need to answer the following questions in details: Growing conditions of the healthy turbots were collected from Tianyuan Aquaculture Co., Ltd., Shandong Provence, China. Age of fish. The pattern of temperature changes is not very clear, especially during recovery (TC group).

Results: I could not find any indication about the data generated in this study being submitted to NCBI or another database, nor could I find this information by searching NCBI. This aspect is crucial to allow in the future retrospective analysis of fastq files. It is the right of the authors to keep the information private until the publication of the manuscript, but please submit them and provide a reviewer’s token in this case. It is crucial for the reproducibility of the analysis. As the most time effective way to publish the data, I would recommend the NCBI GEO database. It allows for the deposition of both raw and processed data (as well as assemblies), processes the data quite rapidly and allows for the generation of a reviewer’s token. On top of that, it would be ideal to also submit the assembly to the TSA / GenBank database, so that it is subsequently included into the databases used by BLAST toolkit.

A detailed list of differentially expressed genes should be provided as an attachment in the article’s supplementary data, so that other researchers can download and understand the detail result of DEGs in the finding. Or consider the option of a heat map with the top 20 or top 30 DEGs in the results.

Discussion:

The discussion is a huge monolithic text. It is separated into logical paragraphs, but it would still be best to separate the subchapters: 4.1.; 4.2. etc. 

I would recommend expanding the discussion by adding a description of some general patterns and providing examples.

1. Most Consistent and Obvious Effect: Impaired Amino Acid Metabolism
Pathway: Valine, leucine, and isoleucine biosynthesis.
Pattern: This is the only pathway that is statistically significantly (low FDR) altered in all three comparison groups. The -ln(p) value is very high in the stress groups (TA and TB) and decreases slightly in the recovery group (TC), but remains significant.
Interpretation: Heat stress causes severe disruptions in the synthesis and catabolism of branched-chain amino acids (BCAAs). These amino acids are critical not only as building blocks for proteins, but also as an alternative energy source (fueling the TCA cycle) and signaling molecules. An imbalance indicates severe energetic and proteostatic stress.

2. Energy Crisis and Impaired Central Metabolism
Pathways associated with energy generation also show a clear pattern:
· Citrate cycle (TCA cycle) and pyruvate metabolism are present in all three groups, but their significance (-ln(p) value) decreases from the TA group to the TC group.
· Glyoxylate and dicarboxylate metabolism are also present in all groups.
· Interpretation: Heat stress (TA, TB) disrupts mitochondrial function and key energy production processes (the Krebs cycle). In the recovery group (TC), these disturbances begin to abate, indicating partial restoration of the body's energy status.

3. Changes in Lipid Metabolism
· Pathways: Glycerolipid metabolism and Glycerophospholipid metabolism. · Pattern: Both pathways are strongly affected by stress (especially Glycerolipid Metabolism in the TA and TC groups). Interestingly, their significance decreases in the TB group (-ln(p) is low), but in the TC group, Glycerolipid Metabolism again becomes one of the most significant pathways.
· Interpretation: Heat stress affects the structure and function of cell membranes (phospholipids) and the metabolism of energy stores (triglycerides). The reappearance of significance in the TC group may indicate active restructuring of membranes and lipid stores during the recovery period.

4. Stress Response and Antioxidant Defense
· Pathway: Glutathione Metabolism
· Pattern: This pathway appears in both stress groups (TA and TB), but disappears from the top 12 in the recovery group (TC). Moreover, its Impact score (influence on the pathway) is very low in all cases. Interpretation: This suggests that glutathione metabolism, a key antioxidant, does respond to stress, but the changes affect only a few metabolites within the pathway (low Impact). The fact that this pathway is no longer significant in the TC group indicates a normalization of the oxidative status after stress relief.

5. Partial Recovery after Stress Relief
Comparison of the TA/TB (stress) groups with the TC (recovery) group shows:
Decreased response intensity: -ln(p) values ​​for most "stress" pathways (BCAA, TCA) are significantly lower in the TC group than in TA and TB.
Emergence of new pathways: Arginine and proline metabolism and pyrimidine metabolism appear in the TC group. This may indicate active recovery processes: arginine and proline are important for protein synthesis and the stress response, while pyrimidines are important for DNA repair and RNA synthesis. · Shift in metabolism: The appearance of the Pentose phosphate pathway in TC may indicate a switch in metabolism to the generation of NADPH for reductive biosyntheses (e.g., synthesis of lipids for membranes).

THUS, heat stress disrupts the fish's central metabolism, causing an energy crisis and disruption of amino acid synthesis. Upon returning to normal conditions, the body begins to recover, as evidenced by a weakening of stress signals and the appearance of metabolic signs of repair and restructuring. However, some disturbances (for example, in BCAA metabolism — Branch-Chein-Amino-Acids metabolism) persist even after the recovery period.

Comments on the Quality of English Language

I'm not a native speaker, but it's obvious the language needs to be edited.

Author Response

Responses to the Reviewer #1’s comments:

Comment 1: Materials and Methods: Experiment Design and Sample Collection are not clearly described. You need to answer the following questions in details: Growing conditions of the healthy turbots were collected from Tianyuan Aquaculture Co., Ltd., Shandong Provence, China. Age of fish. The pattern of temperature changes is not very clear, especially during recovery (TC group).

Response 1: Thanks for your valuable comments. We have added the detail information in the revised manuscript, including the age of fish, the pattern of temperature changes. The growth conditions are the same as the acclimated conditions. Please see 2.1 section, line107, 111-113, 113-121 in the revised manuscript.

Comment 2: Results: I could not find any indication about the data generated in this study being submitted to NCBI or another database, nor could I find this information by searching NCBI. This aspect is crucial to allow in the future retrospective analysis of fastq files. It is the right of the authors to keep the information private until the publication of the manuscript, but please submit them and provide a reviewer’s token in this case. It is crucial for the reproducibility of the analysis. As the most time effective way to publish the data, I would recommend the NCBI GEO database. It allows for the deposition of both raw and processed data (as well as assemblies), processes the data quite rapidly and allows for the generation of a reviewer’s token. On top of that, it would be ideal to also submit the assembly to the TSA / GenBank database, so that it is subsequently included into the databases used by BLAST toolkit.

Response 2: Thanks for your comments. The sequencing data generated in this study are presently not publicly available before the manuscript published, but all the data can be obtained by contacting the corresponding author directly. This has been declared in Data Availability Statement, and all analyzed data are included within the manuscript and supplements. We have checked and added the data in Supplementary materials.

Comment 3: A detailed list of differentially expressed genes should be provided as an attachment in the article’s supplementary data, so that other researchers can download and understand the detail result of DEGs in the finding. Or consider the option of a heat map with the top 20 or top 30 DEGs in the results.

Response 3: Thanks for your comments. We have revised the table in revised manuscript and added the detailed list of DEGs in the Supplementary materials (Table 1-3).

Comment 4: Discussion: The discussion is a huge monolithic text. It is separated into logical paragraphs, but it would still be best to separate the subchapters: 4.1.; 4.2. etc. I would recommend expanding the discussion by adding a description of some general patterns and providing examples.

  1. Most Consistent and Obvious Effect: Impaired Amino Acid Metabolism
    Pathway: Valine, leucine, and isoleucine biosynthesis.
    Pattern: This is the only pathway that is statistically significantly (low FDR) altered in all three comparison groups. The -ln(p) value is very high in the stress groups (TA and TB) and decreases slightly in the recovery group (TC), but remains significant.
    Interpretation: Heat stress causes severe disruptions in the synthesis and catabolism of branched-chain amino acids (BCAAs). These amino acids are critical not only as building blocks for proteins, but also as an alternative energy source (fueling the TCA cycle) and signaling molecules. An imbalance indicates severe energetic and proteostatic stress.
    2. Energy Crisis and Impaired Central Metabolism
    Pathways associated with energy generation also show a clear pattern:
    · Citrate cycle (TCA cycle) and pyruvate metabolism are present in all three groups, but their significance (-ln(p) value) decreases from the TA group to the TC group.
    · Glyoxylate and dicarboxylate metabolism are also present in all groups.
    · Interpretation: Heat stress (TA, TB) disrupts mitochondrial function and key energy production processes (the Krebs cycle). In the recovery group (TC), these disturbances begin to abate, indicating partial restoration of the body's energy status.
    3. Changes in Lipid Metabolism
    · Pathways: Glycerolipid metabolism and Glycerophospholipid metabolism. · Pattern: Both pathways are strongly affected by stress (especially Glycerolipid Metabolism in the TA and TC groups). Interestingly, their significance decreases in the TB group (-ln(p) is low), but in the TC group, Glycerolipid Metabolism again becomes one of the most significant pathways.
    · Interpretation: Heat stress affects the structure and function of cell membranes (phospholipids) and the metabolism of energy stores (triglycerides). The reappearance of significance in the TC group may indicate active restructuring of membranes and lipid stores during the recovery period.
    4. Stress Response and Antioxidant Defense
    · Pathway: Glutathione Metabolism
    · Pattern: This pathway appears in both stress groups (TA and TB), but disappears from the top 12 in the recovery group (TC). Moreover, its Impact score (influence on the pathway) is very low in all cases. Interpretation: This suggests that glutathione metabolism, a key antioxidant, does respond to stress, but the changes affect only a few metabolites within the pathway (low Impact). The fact that this pathway is no longer significant in the TC group indicates a normalization of the oxidative status after stress relief.
    5. Partial Recovery after Stress Relief
    Comparison of the TA/TB (stress) groups with the TC (recovery) group shows:
    Decreased response intensity: -ln(p) values ​​for most "stress" pathways (BCAA, TCA) are significantly lower in the TC group than in TA and TB.
    Emergence of new pathways: Arginine and proline metabolism and pyrimidine metabolism appear in the TC group. This may indicate active recovery processes: arginine and proline are important for protein synthesis and the stress response, while pyrimidines are important for DNA repair and RNA synthesis. · Shift in metabolism: The appearance of the Pentose phosphate pathway in TC may indicate a switch in metabolism to the generation of NADPH for reductive biosyntheses (e.g., synthesis of lipids for membranes).

THUS, heat stress disrupts the fish's central metabolism, causing an energy crisis and disruption of amino acid synthesis. Upon returning to normal conditions, the body begins to recover, as evidenced by a weakening of stress signals and the appearance of metabolic signs of repair and restructuring. However, some disturbances (for example, in BCAA metabolism — Branch-Chein-Amino-Acids metabolism) persist even after the recovery period.

Response 4: Thanks for your valuable comments. In the Discussion section, we have structured our argument by first presenting the expression of differentially expressed genes, followed by their associated signaling pathways, the identified differential metabolites, and finally, an integrated analysis of the transcriptomic and metabolomic data. While we appreciate your suggestion, we believe this structure provides a logical flow that best suits the overall logic of our study. Nevertheless, we have carefully considered all feedback and have revised parts of the Discussion to enhance its clarity and readability. Please see the revised Discussion section in the updated manuscript.

Comment 5: Comments on the Quality of English Language. I'm not a native speaker, but it's obvious the language needs to be edited.

Response 5: Thank you for your valuable comments on our manuscript. We appreciate your attention to the language quality and have carefully addressed this concern. We have engaged a professional English editor to polish the language. Please see the revised manuscript.

Reviewer 2 Report

Comments and Suggestions for Authors

Dear authors,

Please take a look at the attached file.

Best regards

Comments on the Quality of English Language

This manuscript is written in very poor English, making it difficult to understand.

Author Response

Responses to the Reviewer #2’s comments:

Comment 1: Line 54: “Fish are one of constant temperature animals and cannot change their body temperature to adapt to the changes of environment.” This sentence is competent wrong. Fish are cold-blooded vertebrates that regulate their body temperature based on surrounding water typically within plus /minus 0.1 degree Celsius. However, heavy muscled fish such as tuna, have a higher body temperature averaging 10 degree Celsius than their than the surrounding environment.

Response 1: Thanks for pointing this out. We apologize for the error and have corrected the manuscript. Please see the revised manuscript. Please see line 52-56.

“Fish are ectotherms, meaning their body temperature is primarily derived from the external environment. Consequently, the body temperature of most fish species conforms to that of the surrounding water. Their principal strategy for coping with temperature fluctuations is behavioral thermoregulation, such as moving between different water depths or habitats to find optimal temperatures.”

Comment 2: Line 58: “Nile tilapia”

Response 2: Thanks for your valuable comment. We have added the “Nile tilapia”.

Comment 3: Line 92-97: The aim of this study is unclear. Please re-write it clearly.

Response 3: Thanks for your valuable comment. We have thoroughly revised the manuscript to improve clarity. Please see line 97-103 in revised manuscript.

“This study employed an integrated transcriptomic and metabolomic approach to elucidate the molecular mechanisms underlying the thermal stress response in turbot. By analyzing differentially expressed genes (DEGs) and metabolites, we aimed to precisely identify the core regulatory pathways and construct a gene-metabolite interaction network to uncover the systemic adaptive strategy of turbot under thermal stress. Our findings are expected to identify critical candidate genes and metabolic pathways that could serve as potential therapeutic targets for enhancing thermotolerance in turbot.”

Comment 4: In 2.1 section, the experimental design is unclear. Please try to simplify it, or use a figure to simplify it.

Response 4: Thanks for your valuable comment. We have revised the Methods section according to your comments. The description of the temperature rise experiment has been revised. Please see line 113-121.

Comment 5: In 2.4 section. The methodology is unclear and poorly written. Please re-write in clear understandable form.

Response 5: We apologize for the lack of clarity in our initial writing, which was an attempt to reduce textual similarity. Following your suggestion, we have carefully rewritten the section to better clarify. Please see line 146-159.

Comment 6: In 2.7 part. The thermal protocol should be checked.

Response 6: Thanks for your comments. qPCR was performed using the SYBR Green reagent (Cat#R711, Vazyme, China). The experimental procedures were strictly adhered to the manufacturer's instructions. The 2−ΔΔCt method is a well-established and widely accepted approach for gene expression analysis (doi: 10.1016/j.cbd.2019.100632, 10.3390/genes16010009, 10.3390/antiox14010093). We have added relevant citations to substantiate its validity and appropriateness for our study (doi: 10.1016/j.jtherbio.2021.103141, 10.1016/j.aquaculture.2020.735645). Please see line 174-181.

Comment 7: “P” should be written in italics. This figure is unclear due to its very small font size. Please enlarge it.

Response 7: Thanks for your comments. We have revised the manuscript according to your comments. We have uploaded the high-resolution images in revised manuscript.

Comment 8: Figure 6D, which program did the authors used to generate heat map? Please verify. Line 275, what is the InP-value? It is unclear. Please verify

Response 8: Thanks for your valuable comments. We have added the information in the method part. O2PLS was performed on OmicShare platform (https://www.omicshare.com/tools/). (doi: 10.1002/imt2.228). To avoid confusion, we have changed InP-value to log10 P-value.

Reviewer 3 Report

Comments and Suggestions for Authors

Comments to the Author

The manuscript of Xiatian Chen and colleagues is well-written and scientifically valuable study that uses an integrative transcriptome–metabolome approach to elucidate the molecular basis of thermal stress response in juvenile turbot. The study is timely and relevant, considering the impact of climate change on aquaculture. Despite the manuscript's thorough methodology, strong statistical analysis, and concise findings presentation with figures and tables to back up the conclusions, some sections need to be carefully improved for flow, clarity, and scientific rigor before this work can be considered for publishing.

Major comments:

  1. Although the authors list numerous genes, pathways, and metabolites that are differentially expressed, some of their interpretations are overly descriptive. It is important to provide a comprehensive explanation of the biological significance of the FoxO, PPAR, and PI3K-Akt pathways for heat tolerance. Briefly compare your findings with those in other fish species to illustrate distinctiveness and biological importance.
  2. The authors referenced a few earlier studies on turbot in the introduction part, but they did not clarify what is unknown about the molecular pathways or why integrating transcriptomics and metabolomics is so important.
  3. The results contain a lot of data (DEGs tables, metabolite counts, etc.), for example Figures 2, 3, and 4 seems to be dense that may overwhelm readers, therefore keep the key findings (like top DEGs, key pathways, significant metabolites) in the main text and move large tables and detailed lists to the supplementary materials. This will improve the narrative's flow and help readers in focusing on the biological narrative rather than the numerical data.
  4. The conclusion was nicely summarizes the findings however, it could be better to include how these findings could link with the application value like guide selective breeding programs for heat tolerance, inform aquaculture management practices to mitigate thermal stress and serve as a foundation for future functional studies or biomarker development.

Minor comments:

  1. Please explain why you chose 27 °C as the endpoint? Specify if randomization/blinding was used during sample processing.
  2. Please make sure consistent formatting of the gene (italicized) and protein name (not italicized) is used throughout the manuscript.
  3. Add clear figure legends because some readers might struggle to interpret volcano plots and pathway diagrams without referring back to the text.
  4. Provide details on multiple testing corrections for the statistics used in the current study. Please specify the versions of SPSS software used in this work.
  5. Some sections contain grammatical errors and awkward phrasing that need to improve, for example Line 53: With the increasing of global seawater temperature…” and Line 54: “Fish are one of constant temperature animals”. Therefore, it needs careful language editing.
  6. All references should be double-checked to ensure that they follow the journal format.
Comments on the Quality of English Language

The text requires careful language editing.

Author Response

Responses to the Reviewer #3’s comments:

Comment 1: Although the authors list numerous genes, pathways, and metabolites that are differentially expressed, some of their interpretations are overly descriptive. It is important to provide a comprehensive explanation of the biological significance of the FoxO, PPAR, and PI3K-Akt pathways for heat tolerance. Briefly compare your findings with those in other fish species to illustrate distinctiveness and biological importance.

Response 1: Thanks for your valuable comment. We have added the information according to your comments. We have briefly compared those key genes and pathways in fish and discussed the potential effect in turbot under thermal stress. Please see line 361-372, 397-414, 425-427 in revised manuscript.

Comment 2: The authors referenced a few earlier studies on turbot in the introduction part, but they did not clarify what is unknown about the molecular pathways or why integrating transcriptomics and metabolomics is so important.

Response 2: Thanks for your valuable comment. We have added the information according to your comments. Previous studies have primarily focused on phenotypic and isolated molecular changes, lacking a systems-level understanding. The section on multi-omics methodology was revised to highlight the critical advantage of integrated analysis in closing the gap between genetic potential and phenotypic manifestation, thereby justifying its necessity for this study. Please see line 78-84, 93-103 in the Introduction section of revised manuscript.

Comment 3: The results contain a lot of data (DEGs tables, metabolite counts, etc.), for example Figures 2, 3, and 4 seems to be dense that may overwhelm readers, therefore keep the key findings (like top DEGs, key pathways, significant metabolites) in the main text and move large tables and detailed lists to the supplementary materials. This will improve the narrative's flow and help readers in focusing on the biological narrative rather than the numerical data.

Response 3: Thanks for your valuable comment. We have revised the Result section, and deleted the content of tables. The detailed lists were moved to the Supplementary materials.

Comment 4: The conclusion was nicely summarizes the findings however, it could be better to include how these findings could link with the application value like guide selective breeding programs for heat tolerance, inform aquaculture management practices to mitigate thermal stress and serve as a foundation for future functional studies or biomarker development.

Response 4: Thanks for your valuable comment. We have added our findings' practical value and the future applications. Please see line 437-454 in revised manuscript.

Comment 5: Please explain why you chose 27 °C as the endpoint? Specify if randomization/blinding was used during sample processing.

Response 5: Thanks for your comment. In our previous study (doi: 10.1016/j.aquaculture.2020.735645), we found that juvenile turbot (100–120 g) began to experience mortality after being acclimated and then exposed to a gradual temperature increase from 18 °C to 27 °C, maintained at 27 °C for five days. This experimental condition has been described in the Methods section. The fish were randomly divided into different groups, and samples were collected accordingly. Please see line 106-121 in revised manuscript.

Comment 6: Please make sure consistent formatting of the gene (italicized) and protein name (not italicized) is used throughout the manuscript.

Response 6: Thanks for your comment. We have revised the manuscript according to your comment.

Comment 7: Add clear figure legends because some readers might struggle to interpret volcano plots and pathway diagrams without referring back to the text.

Response 7: Thanks for your comment. We have revised the figure legends according to your comment.

Comment 8: Provide details on multiple testing corrections for the statistics used in the current study. Please specify the versions of SPSS software used in this work.

Response 8: Thanks for your comment. We have revised the manuscript according to your comment. Please see line 186-191 in revised manuscript.

Comment 9: Some sections contain grammatical errors and awkward phrasing that need to improve, for example Line 53: With the increasing of global seawater temperature…” and Line 54: “Fish are one of constant temperature animals”. Therefore, it needs careful language editing. The text requires careful language editing.

Response 9: Thank you for your valuable comments on our manuscript. We appreciate your attention to the language quality and have carefully addressed this concern. We have engaged a professional English editor to polish the language.

Comment 10: All references should be double-checked to ensure that they follow the journal format.

Response 10: Thanks for your comment. We have revised the manuscript based on your comment and checked the references style.

Reviewer 4 Report

Comments and Suggestions for Authors

Summary:

This study by Chen et al. investigates the molecular mechanisms of thermal stress in juvenile turbot (Scophthalmus maximus) by integrating transcriptomic and metabolomic profiling. The authors identify thousands of differentially expressed genes (DEGs) and dozens of differentially regulated metabolites, linking them to amino acid metabolism, steroid biosynthesis, PI3K/Akt, PPAR, and FoxO signaling. Notably, correlations are reported between genes (e.g., CISH, Plch2, DNAJB6) and metabolites such as valine, leucine, galactonic acid, and α-santonin-1. The multi-omics design is a major strength, providing novel insights into heat stress biology with potential applications in aquaculture, though several aspects require clarification.

Major comments:

  1. The repeated emphasis on α-santonin-1 as a metabolite of interest warrants clarification. α-Santonin is a sesquiterpene lactone naturally occurring in Artemisia species and historically used as an anthelmintic in humans, but it is not known to be an endogenous vertebrate metabolite. In untargeted metabolomics, compound names often derive from automated database matches (e.g., KEGG, PubChem), and suffixes such as “-1” may indicate isomers or database artifacts rather than validated structures. Indeed, misannotation is a well-recognized limitation of untargeted metabolomics, as many features are matched based only on accurate mass rather than confirmed fragmentation spectra. Without confirmation by authentic standards or MS/MS spectral matching, the detection of α-santonin-1 in turbot liver is more likely an annotation artifact or feed-derived exogenous compound than a genuine metabolic product. The authors need to temper their conclusions regarding α-santonin-1 and clearly state the confidence level of metabolite identification, in line with best practices for metabolomics reporting.

  2. As the fish were obtained from a food processing company, the authors should state the limitation that these animals might be genetically diverse (not sibling controls) in the paper.

  3. Were transcriptomics and metabolomics analyses performed on the same liver samples collected from the same animals, or do the samples originate from different cohorts? The difference could affect the “association analysis of metabolome and transcriptome.” Please clarify.

  4. The discussion of SET-1 upregulation would benefit from stronger context on epigenetic regulation in fish. SET-1 belongs to the SET-domain histone methyltransferases that modify H3 and H4 residues, influencing chromatin structure and transcription. Epigenetic responses are increasingly recognized as important in fish stress biology, including heat-induced changes in DNA methylation and histone modification in zebrafish, salmon, and other aquaculture species. The authors should expand on how SET-1 expression may be linked to these processes in turbot, while acknowledging that their findings represent associations, not mechanistic proof. Including references to fish epigenetics and outlining possible follow-up approaches (e.g., ChIP-seq, DNA methylation assays) would strengthen this section and better frame the conclusions.”

Minor comments:

  1. The simple summary and the abstract are remarkably similar. Ideally, the simple summary should be tailored more towards a general audience. Perhaps not mentioning any gene names, but what processes they control and more importantly, more explicitly stating why the altered pathways are important would be more valuable. By looking at the simple summary section, the general audience should be able to understand why altered “PI3K/Akt signaling pathway, protein transport, and protein processing” is important – both to the organism and to us.

  2. Figure 3A-C: For GO analyses, having the process id alone is not informative. Please provide the pathway/process/molecular function name on the figure. Otherwise, these panels can be completely removed, and the reader can rely on 3D-F for pathway analyses.

  3. Lines 222-228: While I appreciate why these analyses were performed, it is unclear what information they ultimately provided here. For example, the authors write “The DEGs in CTL vs TB group were mainly involved in cellular process, biological regulation, metabolic process and response to stimulus in BP category”. Results like these are overly general and do not add value to the comparison between conditions.

  4. Figure 4A: Please use the same sample abbreviations for this panel as the ones used throughout the manuscript.

  5. Figure 4C-D: The volcano plots could be improved by calling out the most significantly upregulated and downregulated genes on the figure.

Comments on the Quality of English Language

Please do a careful reading of the manuscript to work out grammatical issues, e.g. “Take appropriate weight liver sample into the 1.5 ml centrifuge tubes, added appropriate volume L-2-chlorophenylalanine and methanol.” (Lines 140-141)

Author Response

Responses to the Reviewer #4’s comments:

Comment 1: The repeated emphasis on α-santonin-1 as a metabolite of interest warrants clarification. α-Santonin is a sesquiterpene lactone naturally occurring in Artemisia species and historically used as an anthelmintic in humans, but it is not known to be an endogenous vertebrate metabolite. In untargeted metabolomics, compound names often derive from automated database matches (e.g., KEGG, PubChem), and suffixes such as “-1” may indicate isomers or database artifacts rather than validated structures. Indeed, misannotation is a well-recognized limitation of untargeted metabolomics, as many features are matched based only on accurate mass rather than confirmed fragmentation spectra. Without confirmation by authentic standards or MS/MS spectral matching, the detection of α-santonin-1 in turbot liver is more likely an annotation artifact or feed-derived exogenous compound than a genuine metabolic product. The authors need to temper their conclusions regarding α-santonin-1 and clearly state the confidence level of metabolite identification, in line with best practices for metabolomics reporting.

Response 1: Thanks for your valuable comment. In this study, we detected a signal corresponding to α-santonin-1 in the liver metabolome of turbot. It is important to note that the identification of compounds in non-targeted metabolomics relies on mass spectral matching against standard databases, and many endogenous metabolites may share similar mass-to-charge ratios and fragmentation patterns. Our non-targeted metabolomic analysis was performed using a gas chromatography-mass spectrometry system (GeneDenovo Biotechnology, Guangzhou, China). Data acquisition was carried out with Chroma TOF 4.3X software (LECO), and metabolites were filtered using a quality control criterion of relative standard deviation (RSD) < 30% in QC samples, which ensures the analytical reliability of the detected signals.

Therefore, it is possible that this signal corresponds to an as-yet-unidentified endogenous metabolite with chemical properties similar to those of α-santonin. Alternatively, the compound may have originated from plant-based ingredients in the experimental diet and accumulated in the liver via dietary intake. We consider this signal as a candidate finding that requires further validation. We have revised the manuscript according to your comment.

Comment 2: As the fish were obtained from a food processing company, the authors should state the limitation that these animals might be genetically diverse (not sibling controls) in the paper.

Response 2: Thanks for your valuable comment. We have added this limitation in the revised manuscript. Please see line 450-454.

Comment 3: Were transcriptomics and metabolomics analyses performed on the same liver samples collected from the same animals, or do the samples originate from different cohorts? The difference could affect the “association analysis of metabolome and transcriptome.” Please clarify.

Response 3: Thanks for your valuable comment. We have revised the Methods section to explicitly state that the metabolomic and transcriptomic analyses were performed on the same liver samples collected from the same set of fish. Please see method section, line 146-147.

Comment 4: The discussion of SET-1 upregulation would benefit from stronger context on epigenetic regulation in fish. SET-1 belongs to the SET-domain histone methyltransferases that modify H3 and H4 residues, influencing chromatin structure and transcription. Epigenetic responses are increasingly recognized as important in fish stress biology, including heat-induced changes in DNA methylation and histone modification in zebrafish, salmon, and other aquaculture species. The authors should expand on how SET-1 expression may be linked to these processes in turbot, while acknowledging that their findings represent associations, not mechanistic proof. Including references to fish epigenetics and outlining possible follow-up approaches (e.g., ChIP-seq, DNA methylation assays) would strengthen this section and better frame the conclusions.”

Response 4: Thanks for your valuable comment. We have added this section according to your comments. We discussed the association between SET-1, epigenetic regulation and thermal stress. Please see line 357-372 in revised manuscript.

Comment 5: The simple summary and the abstract are remarkably similar. Ideally, the simple summary should be tailored more towards a general audience. Perhaps not mentioning any gene names, but what processes they control and more importantly, more explicitly stating why the altered pathways are important would be more valuable. By looking at the simple summary section, the general audience should be able to understand why altered “PI3K/Akt signaling pathway, protein transport, and protein processing” is important – both to the organism and to us.

Response 5: Thanks for your valuable com14-27 in revised manuscript.

Comment 6: Figure 3A-C: For GO analyses, having the process id alone is not informative. Please provide the pathway/process/molecular function name on the figure. Otherwise, these panels can be completely removed, and the reader can rely on 3D-F for pathway analyses.

Response 6: Thanks for your valuable comment. We described the significant GO terms of Figure 3A-3C in the Result section (line 229-243), and the detailed list containing the GO IDs, their corresponding full names, enrichment scores, and p-values has been included as supplementary table 5-6. The chord plot could create a clear visual distinction from the KEGG pathway analysis presented in panels D-F. We have uploaded the high resolution figures in the revised manuscript.

Comment 7: Lines 222-228: While I appreciate why these analyses were performed, it is unclear what information they ultimately provided here. For example, the authors write “The DEGs in CTL vs TB group were mainly involved in cellular process, biological regulation, metabolic process and response to stimulus in BP category”. Results like these are overly general and do not add value to the comparison between conditions.

Response 7: Thanks for your comment. The purpose of presenting the GO analysis in this section was to provide a systematic overview of the functional distribution of DEGs across three fundamental categories: BP, CC, and MF. This approach is widely used in transcriptomic studies to initially characterize the global functional attributes of DEGs (doi: 10.1016/j.jhazmat.2021.125862; 10.1016/j.ijbiomac.2025.139542). In this study, the GO terms enriched by the DEGs showed considerable similarity. While subtle differences exist, the overarching functional themes—such as cellular processes, metabolic activities, and stimulus response—were consistently observed. We have checked and revised the figure and legends according to the feedback.

Comment 8: Figure 4A: Please use the same sample abbreviations for this panel as the ones used throughout the manuscript.

Response 8: Thanks for your valuable comment. We have revised figure 4A according to your comment.

Comment 9: Figure 4C-D: The volcano plots could be improved by calling out the most significantly upregulated and downregulated genes on the figure.

Response 9: Thank for your comment. In response to the collective feedback from all reviewers regarding the clarity of result presentation, we have taken the following actions: To prevent overcrowding in the volcano plots, we deliberately did not label all data points in Figure 4B-D, as the key findings from these figures are described in detail in the main text. Furthermore, we have carefully reviewed Figure 2 and updated its corresponding table to ensure that readers can easily locate the differentially expressed genes and metabolites.

Comment 10: Please do a careful reading of the manuscript to work out grammatical issues, e.g. “Take appropriate weight liver sample into the 1.5 ml centrifuge tubes, added appropriate volume L-2-chlorophenylalanine and methanol.” (Lines 140-141)

Response 10: We are grateful for these valuable comments. Accordingly, we have thoroughly revised the manuscript to incorporate more precise methodological descriptions. The manuscript has undergone professional language editing by a native speaker to ensure the clarity and accuracy of the presentation.

Round 2

Reviewer 1 Report

Comments and Suggestions for Authors

The article has been substantially revised by the authors.
The Simple Summary does not duplicate the Abstract.
Substantial additions have been made to Section 1. Introduction, emphasizing the importance and clarifying the objective of the study.
Section 2. Materials and Methods has been substantially revised.
The updated "Discussion" section in the revised manuscript is truly more complete and easier to read.
Regarding the sequencing data, this is the first time I've encountered this type of data placement. In fact, the original materials presented in the study are NOT included in the article! The primary sequencing data itself is valuable for other researchers and should be deposited in a database. Therefore, the article can be published in its current form, but I leave this decision to the editor's discretion.

Therefore, after revision by the authors, the article can be published, but with minor revisions.

Minor comments:
2. Materials and Methods
In Section 2.1. It's not specified how many fish were selected for sequencing and how many for metabolomic analysis, and what happened to these samples next? Were they pooled or...?
Judging by the subsequent presentation of the results in the illustrative material, it's likely that these were:
1) 3 fish/samples from each group for sequencing (or perhaps each sample was a pool of 3 fish, and a total of 3 pools/samples for each group were analyzed)?
2) 10 fish/samples for metabolomic analysis?

In Section 2.7., were the RNA used for qPCR analysis collected from the same individuals as those used for transcriptomic sequencing?

3. Results
In fact, after adding Tables 1-3 to the supplementary section, Tables 3-5 in the main text duplicate the appendix. It seems logical to present these results in a more visual format, where all three tables would be displayed in a single figure  (as a heat map instead of 3 tables).

Section 3.6, line 300 - is it 28 or 27 metabolites?
Section 3.7, line 333 - it seems that  the expression of the DNAJB6 gene in the restoration group differ from the control?

4. Discussion
Line 348: A total of 5,413 genes were dysregulated in fish under heat stress, including 127 or 126 common genes?

Author Response

Comment 1:Regarding the sequencing data, this is the first time I've encountered this type of data placement. In fact, the original materials presented in the study are NOT included in the article! The primary sequencing data itself is valuable for other researchers and should be deposited in a database. Therefore, the article can be published in its current form, but I leave this decision to the editor's discretion.

Response 1:The data used to support the findings of this study have been deposited in the NCBI’s Sequence Read Archive (SRA), reference number (BioProject ID: PRJNA1344168). We have added the statement in the revised manuscript. Please see line 484-488 in revised manuscript.

Comment 2: 2. Materials and Methods. In Section 2.1. It's not specified how many fish were selected for sequencing and how many for metabolomic analysis, and what happened to these samples next? Were they pooled or...?

Judging by the subsequent presentation of the results in the illustrative material, it's likely that these were: 1) 3 fish/samples from each group for sequencing (or perhaps each sample was a pool of 3 fish, and a total of 3 pools/samples for each group were analyzed)? 2) 10 fish/samples for metabolomic analysis?

Response 2: Thanks for your comments. We have clarified that the fish used in this study were randomly selected. A total of 120 fish were randomly assigned to four experimental groups, each with three biological replicates. For both transcriptomic and metabolomic analyses, three fish per group were randomly chosen, as illustrated in Result section. This experimental design is consistent with methodologies employed in previous studies (doi: 10.1016/j.cbd.2019.100632; 10.1016/j.aquaculture.2019.734830).

Regarding metabolomic analysis, as described in Section 2.4, “The liver samples used for this metabolomic analysis were collected from the same individuals as those used for transcriptomic profiling.”

Comment 3: In Section 2.7., were the RNA used for qPCR analysis collected from the same individuals as those used for transcriptomic sequencing?

Response 3: Thanks for your comment. We confirm that the qPCR validation was performed not only on the same fish used for transcriptome sequencing, but also included additional fish subjected to the same experimental conditions. This expanded sampling approach enhances the robustness and reliability of our gene expression data, as reflected in the results presented in Figure 7. All relevant qPCR data have been made publicly available via the Zenodo repository (doi: 10.5281/zenodo.17299068).

Comment 4: 3. Results. In fact, after adding Tables 1-3 to the supplementary section, Tables 3-5 in the main text duplicate the appendix. It seems logical to present these results in a more visual format, where all three tables would be displayed in a single figure (as a heat map instead of 3 tables).

Response 4: Thanks for your comment. We have opted to retain Tables 3–5 as separate entities rather than consolidating all differentially expressed gene information into a single table. This structure is intended to enhance clarity and facilitate a more organized and accessible presentation of the results, allowing readers to better interpret the dataset in a structured manner.

Comment 5: Section 3.6, line 300 - is it 28 or 27 metabolites?

Response 5: We are grateful for your careful review and valuable feedback. We have corrected the error and have conducted a comprehensive check of all numerical values throughout the manuscript to prevent similar issues.

Comment 6: Section 3.7, line 333 - it seems that the expression of the DNAJB6 gene in the restoration group differ from the control?

Response 6: Thanks for your comment. The expression levels of not only DNAJB6, but also slc25a25a and CYP7A1, in the recovery group were found to differ from those in the control group. These differences are described in the Results section.

Comment 7: 4. Discussion. Line 348: A total of 5,413 genes were dysregulated in fish under heat stress, including 127 or 126 common genes?

Response 7: Thanks for your comment. We have revised the manuscript.

Reviewer 2 Report

Comments and Suggestions for Authors

Dear author,

Thank you for your effort.

Author Response

Comment: Dear author, Thank you for your effort.

Response: We sincerely thank you for your insightful suggestions throughout the review process.

Reviewer 3 Report

Comments and Suggestions for Authors

The manuscript has been greatly improved, and I am satisfied with the revisions. I have no further comments or suggestions at this point. My best wishes

Author Response

Comment: The manuscript has been greatly improved, and I am satisfied with the revisions. I have no further comments or suggestions at this point. 

Response: We sincerely thank you for your insightful suggestions throughout the review process.

Reviewer 4 Report

Comments and Suggestions for Authors

Summary:

I would like to thank the authors for their thorough and thoughtful revisions. The manuscript has improved substantially in clarity, structure, and contextual depth. The expanded discussion of SET-1 and epigenetic regulation is particularly well done, and the revisions to the Simple Summary, the addition of the genetic diversity limitation, and the clarification that transcriptomic and metabolomic analyses were conducted on the same individuals have all strengthened the paper.

However, several important issues remain unresolved. Most notably, the handling of α-santonin-1 continues to overstate its identification — it is still presented as a confirmed endogenous metabolite without qualification or discussion of annotation confidence, despite the lack of MS/MS validation. Similarly, Figure 3A–C remain visually unclear and uninformative without GO term labels, and the accompanying text continues to provide only generic descriptions with limited biological interpretation. These points require further attention before the manuscript can be considered fully satisfactory.

Overall, the study is promising and now much improved, but a few targeted revisions, particularly clarification of metabolite identification and refinement of the GO analysis presentation, are still needed to ensure rigor and interpretability.

Specific comments: 

1) While the authors acknowledge in their response that α-santonin-1 may represent either an unidentified endogenous metabolite or a plant-derived exogenous compound, the revised manuscript does not actually reflect this clarification. In the updated version, α-santonin-1 remains presented throughout the Results, Discussion, and Conclusion sections as a bona fide endogenous metabolite. No statement of metabolite identification confidence level, no mention of MS/MS spectral confirmation, and no explicit cautionary note regarding potential misannotation have been added. Because untargeted GC–MS identifications based solely on database spectral matching are inherently ambiguous, this metabolite must be clearly described as putatively annotated and discussed as tentative until validated with authentic standards or targeted MS/MS. The authors should temper their interpretation throughout and revise the manuscript to reflect that α-santonin-1 is an uncertain or exogenous feature, not confirmed evidence of an endogenous metabolic response to heat stress. As written, this issue remains unresolved.

2) The authors have expanded the Results section to describe the major GO categories and have added full term names and statistics to the supplementary tables, which improves interpretability. However, Figure 3A–C themselves still do not include GO term names, showing only broad category plots (BP, CC, MF) without specific functional terms. As a result, the figures remain visually appealing but not self-explanatory; readers must cross-reference the supplementary tables to understand what each GO segment represents.

If the authors intend to keep these panels, I recommend adding at least the top 5–10 enriched GO term names (or representative process labels) directly onto the plots or beside them in the legend. Alternatively, the authors could simplify the figure set by omitting panels A–C and referring readers to the supplementary tables, as the KEGG pathway panels already convey the more biologically informative results.

3) Although the authors explained their rationale for including GO summaries, they did not improve the scientific content of this section. The text continues to report only generic GO categories without drawing comparisons between groups or identifying any biologically meaningful differences. These statements add little interpretive value and could be streamlined or moved entirely to Supplementary Materials. I maintain that this section remains overly general and should either (i) highlight specific biological themes that differ across treatments or (ii) be condensed to avoid redundancy with KEGG enrichment.

Author Response

Comment 1: While the authors acknowledge in their response that α-santonin-1 may represent either an unidentified endogenous metabolite or a plant-derived exogenous compound, the revised manuscript does not actually reflect this clarification. In the updated version, α-santonin-1 remains presented throughout the Results, Discussion, and Conclusion sections as a bona fide endogenous metabolite. No statement of metabolite identification confidence level, no mention of MS/MS spectral confirmation, and no explicit cautionary note regarding potential misannotation have been added. Because untargeted GC–MS identifications based solely on database spectral matching are inherently ambiguous, this metabolite must be clearly described as putatively annotated and discussed as tentative until validated with authentic standards or targeted MS/MS. The authors should temper their interpretation throughout and revise the manuscript to reflect that α-santonin-1 is an uncertain or exogenous feature, not confirmed evidence of an endogenous metabolic response to heat stress. As written, this issue remains unresolved.

Response 1: Thanks for your comment. We have revised the Results, Discussion, and Conclusion sections. We have added appropriate cautionary statements in the Results section to highlight this uncertainty according to your comment. Our metabolic analysis was conducted following standard protocols, and we have presented the complete results without selective editing. These modifications ensure a more accurate representation of our findings while maintaining the integrity of our original data interpretation. The limitations of this interpretation and the necessity for confirmatory studies have been explicitly discussed in our manuscript. Please see line 273-280, 299-303, 312-313,448 in revised manuscript.

Comment 2: The authors have expanded the Results section to describe the major GO categories and have added full term names and statistics to the supplementary tables, which improves interpretability. However, Figure 3A–C themselves still do not include GO term names, showing only broad category plots (BP, CC, MF) without specific functional terms. As a result, the figures remain visually appealing but not self-explanatory; readers must cross-reference the supplementary tables to understand what each GO segment represents.

If the authors intend to keep these panels, I recommend adding at least the top 5–10 enriched GO term names (or representative process labels) directly onto the plots or beside them in the legend. Alternatively, the authors could simplify the figure set by omitting panels A–C and referring readers to the supplementary tables, as the KEGG pathway panels already convey the more biologically informative results.

Response 2: Thanks for your comments, we have revised Figure 3 according to your comment. Please see Figure 3 and Result section, 3.3 part in revised manuscript.

Comment 3: Although the authors explained their rationale for including GO summaries, they did not improve the scientific content of this section. The text continues to report only generic GO categories without drawing comparisons between groups or identifying any biologically meaningful differences. These statements add little interpretive value and could be streamlined or moved entirely to Supplementary Materials. I maintain that this section remains overly general and should either (i) highlight specific biological themes that differ across treatments or (ii) be condensed to avoid redundancy with KEGG enrichment.

Response 3: Thanks for your comment. We have revised the manuscript according to your comment. Regarding the GO analysis, the results, supplementary figures, and the updated Figure 3 now provide a comprehensive presentation of the findings. As the turbot in the high-temperature groups exhibited similar GO enrichment profiles, we have concisely highlighted the specific biological themes that differed significantly across treatments in the Results section.